# Analysis of aPTT predictors after unfractionated heparin administration in intensive care units using machine learning models

Tadashi Kamio[1,2]*, Masaru Ikegami[3], Megumi Mizuno[3], Seiichiro Ishii[3], Hayato Tajima[3], Yoshihito Machida[3], Kiyomitsu Fukaguchi[2]

1 Department of Anesthesiology and Critical Care Medicine, Jichi Medical University Saitama Medical Center, Saitama, Japan, 2 Department of Critical Care Medicine, Shonan Kamakura General Hospital, Kamakura, Kanagawa, Japan, 3 Terumo Corporation, Shonan Center, Ashigarakami-gun, Kanagawa, Japan

* tadashi-kamio@mail.goo.ne.jp

## Abstract

### Objectives

Predicting optimal coagulation control using heparin in intensive care units (ICUs) remains a significant challenge. This study aimed to develop a machine learning (ML) model to predict activated partial thromboplastin time (aPTT) in ICU patients receiving unfractionated heparin for anticoagulation and to identify key predictive factors.

### Methods

Data were obtained from the Tokushukai Medical Database, covering six hospitals with ICUs in Japan, collected between 2018 and 2022. The study included 945 ICU patients who received unfractionated heparin. The dataset comprised both static and dynamic features, which were used to construct and train ML models. Models were developed to predict aPTT following initial and multiple heparin doses. Model performance was evaluated using the area under the receiver operating characteristic curve (ROC AUC), area under the precision–recall curve (PR AUC), precision, recall, F1 score, and accuracy. SHAP analysis was conducted to determine key predictive factors.

### Results

The random forest model demonstrated the highest predictive performance, with ROC AUC values of 0.707 for the first infusion and 0.732 for multiple infusions. Corresponding PR AUC values were 0.539 and 0.551. Despite moderate overall predictive performance, the model exhibited high precision (0.585 for the first infusion and 0.589 for multiple infusions), indicating effectiveness in correctly identifying true positive cases. However, recall and F1 scores were lower, suggesting that some

**Data availability statement:** Data availability statement This study was conducted using patient data obtained from the Tokushukai Medical Database. The dataset includes clinical information necessary for predicting activated partial thromboplastin time (aPTT) in intensive care unit (ICU) patients receiving unfraction-ated heparin (UFH). Although anonymization procedures have been applied, the Tokushukai Medical Database is an integrated electronic health record system managed by a third-party organization, Tokushukai Information System Inc.. Consequently, the authors do not have the authority to publicly share the dataset. Additionally, data were collected from six hos-pitals, and according to the ethical guidelines of each participating institution, external data sharing is strictly prohibited. Institutional Review Boards Imposing Data Restrictions This study was approved by the following Institutional Review Boards (IRBs): Tokushukai Group Joint Ethics Review Committee (Approval Number: TGE02078-024), Terumo Research Ethics Review Committee (Processing Number: CR22-R052) The study was con-ducted in accordance with the Declaration of Helsinki, and the opt-out consent procedure was approved by these ethics committees, waiving the requirement for individual patient consent. However, secondary use or external provision of the dataset is restricted. Due to ethical restrictions, the dataset used in this study is not publicly available. For inquiries regarding access to the dataset, please contact the Tokushukai Group Joint Ethics Review Committee directly: Tokushukai Group Joint Ethics Review Committee (URL: https://www.tokushukai.or.jp/research/database/)

**Funding:** The author(s) received no specific funding for this work.

**Competing interests:** The authors have declared that no competing interests exist.

cases, particularly in sub-therapeutic and supra-therapeutic ranges, may have been missed. Incorporating time-series data, such as vital signs, provided only marginal improvements in performance.

## Conclusions

ML models demonstrated moderate performance in predicting aPTT following heparin infusion in ICU patients, with the random forest model achieving the highest classifi-cation accuracy. Although the models effectively identified true positive cases, their overall predictive performance remained limited, necessitating further refinement. The inclusion of static and dynamic features did not significantly enhance model accuracy. Future studies should explore additional factors to improve predictive models for optimizing individualized anticoagulation management in ICUs.

## Introduction

Unfractionated heparin (UFH) is widely used as an anticoagulant in intensive care units (ICUs); however, its poor controllability renders precise dosing challenging, increasing the risk of complications such as bleeding and thrombosis. These risks vary based on physician dosing decisions. Clinicians typically initiate UFH therapy with a bolus dose for immediate anticoagulation, followed by a weight-based contin-uous infusion (IU/kg/h). However, fewer than 50% of patients consistently achieve therapeutic activated partial thromboplastin times (aPTTs) after starting UFH therapy, underscoring the need for more precise dosing strategies [1,2].

In recent years, machine learning (ML) approaches have been increasingly applied in healthcare to predict treatment outcomes and assess clinical risks. Electronic health records (EHRs), which contain diagnostic information, medication histories, vital signs, and laboratory results, are valuable resources for developing predictive models. ML-based models trained on EHR data can support clinicians in making earlier and more informed decisions, offering a promising tool for improving anticoag-ulation management. Several studies have explored ML-based models for predicting UFH dosing and aPTT responses using algorithms such as logistic regression, deci-sion trees, and recurrent neural networks (RNNs) [3–5].

For instance, Ghassemi et al. developed a data-driven approach to optimize heparin dosing; however, their model was limited by incomplete data on initial bolus doses and significant variability in heparin administration patterns. Similarly, Su et al. applied ML techniques to compare multiple heparin dosing models using the MIMIC-III database [4]. Although their model showed promise, challenges remained in handling continuous variables such as the Sequential Organ Failure Assessment (SOFA) score and the cumulative impact of heparin administration. In contrast, Boie et al. demonstrated improved aPTT prediction by incorporat-ing time-series data into an RNN-based ML model [6]. Although their approach enhanced predictive accuracy, it was based on a single-center dataset, limiting its

generalizability. To date, no study has validated these models using multicenter data, leaving their applicability across diverse hospital settings uncertain.

Moreover, critical gaps persist in ML-based UFH dosing research. A recent systematic review identified only a limited number of studies specifically addressing ML models for UFH dosing, and none of the proposed models have been integrated into routine clinical practice [7]. Additionally, most prior studies have relied on single-center datasets or performed internal validation exclusively within publicly available databases such as MIMIC-III and eICU, raising concerns about generalizability [3–6] .Furthermore, while existing models primarily predict aPTT following initial heparin administration, no studies have focused on predicting aPTT following multiple does—an essential consideration for clinical application. Moreover, most models have been developed using only static variables, overlooking the importance of dynamic factors that are crucial for real-time clinical decision-making [8].

This study aims to address these gaps by developing and validating an ML model that integrates both static patient characteristics and dynamic treatment–response variables. In contrast to previous models, our approach leverages a large, multicenter database from Japan, enhancing its generalizability and clinical applicability. We hypothesize that incorporating both static features (e.g., age and blood test results) and dynamic variables (e.g., vital signs) will improve predictive accuracy and support real-time anticoagulation management in ICU settings.

## Method

### Data source

We conducted a retrospective cohort study using data from the Tokushukai Medical Database, covering the period from January 1, 2018, to June 30, 2022. The Tokushukai Medical Group, Japan's largest private medical service provider, operates 77 hospitals nationwide (https://www.tokushukai.or.jp/en/). The database, managed by Tokushukai Information System Inc., consolidates EHRs from each hospital, utilizing standardized medical coding for drug names and laboratory values. This structure enables the extraction of patient data from individual hospital registration systems. The database is frequently used in medical research, and multiple studies based on this resource have been published [9–11].

For this study, we retrospectively analyzed data from six hospitals with ICUs, all of which participate in the Diagnosis Procedure Combination system [12]. The database provides comprehensive patient information, including demographics (age and gender), vital signs, all in-hospital laboratory measurements, admission and discharge dates, discharge outcomes (alive or deceased), primary admission diagnoses, comorbidities recorded at discharge using International Classification of Diseases, 10th Revision (ICD-10) codes, comorbidities documented at admission, post-admission complications, medications, and procedures such as acute dialysis and extracorporeal membrane oxygenation (ECMO) [13].

This study was approved by the Tokushukai Group Joint Ethics Review Committee (Approval No.: TGE02078−024) and the Terumo Research Ethics Review Committee (Processing No.: CR22-R052) and was conducted in accordance with the principles of the Declaration of Helsinki. The Ethics Review Committees approved an opt-out consent process, waiving the requirement for individual consent. Notifications were provided to participants via the Tokushukai Group's website (https://www.tokushukai.or.jp/research/database/). Data for this retrospective study were accessed on January 10, 2023, for research purposes. The authors did not have access to any information that could identify individual participants during or after data collection.

### Study participants

The study included adult patients (≥18 years) who were admitted on an emergency or unscheduled basis to six hospitals between January 1, 2018, and June 30, 2022, and received heparin therapy (either bolus or continuous) in the ICUs. Patients admitted to emergency care units (ECUs) were also included, as ECUs in Japan function similarly to ICUs, with no clear distinction between them.

Patients were excluded if they lacked aPTT measurements either before or after heparin administration. Specifically, exclusion criteria included cases where the target aPTT was not measured within 6–24 h following heparin administration, heparin dosing was completed before the target aPTT was measured, or the reference aPTT was not recorded during the predictor variable period.

In ICUs, heparin was administered multiple times to individual patients. Each administration event was treated as an independent data entry, leading to multiple datasets for some patients (Fig 1). Detailed exclusion criteria and examples are provided in S1 Table in S2 File. The recorded time of administration after aPTT measurement was considered the starting point for all bolus and continuous heparin administrations.

## Classification of heparin therapeutic effects

We categorized patients into sub-therapeutic, normal-therapeutic, and supra-therapeutic groups based on their aPTT values following heparin treatment. In Japanese clinical practice, aPTT test results typically become available approximately 6 h after the initiation of heparin administration. Therefore, the therapeutic aPTT value was defined as the first aPTT measurement obtained between 6 and 24 h after the initial heparin infusion. Due to significant variability in aPTT distributions resulting from differences in patient characteristics and treatment regimens, we conducted separate analyses and developed predictive models for both initial and multiple heparin administration cases. The clinical outcome prediction task was framed as a ternary classification problem. In clinical practice, the normal therapeutic range is generally set at 1.5 to 2 times the patient's reference value. If ECMO is used, this value is doubled. Thus, given that the reference aPTT is 40 s, and based on thresholds used in previous studies, we classified patients into the following groups: sub-therapeutic (<40 s), normal-therapeutic (40–80 s), and supra-therapeutic (>80 s) [3].

## Predictor variables

We extracted 46 variables from the database that were expected to predict the study outcomes. These variables included demographic information (gender, age, height, weight, and blood type), physiological parameters (circulating blood volume [CBV], body temperature, respiratory rate, pulse rate, $SpO_2$, systolic blood pressure [SBP], and diastolic blood pressure [DBP]), clinical history (underlying disease, surgery, extracorporeal membrane oxygenation [ECMO], continuous renal

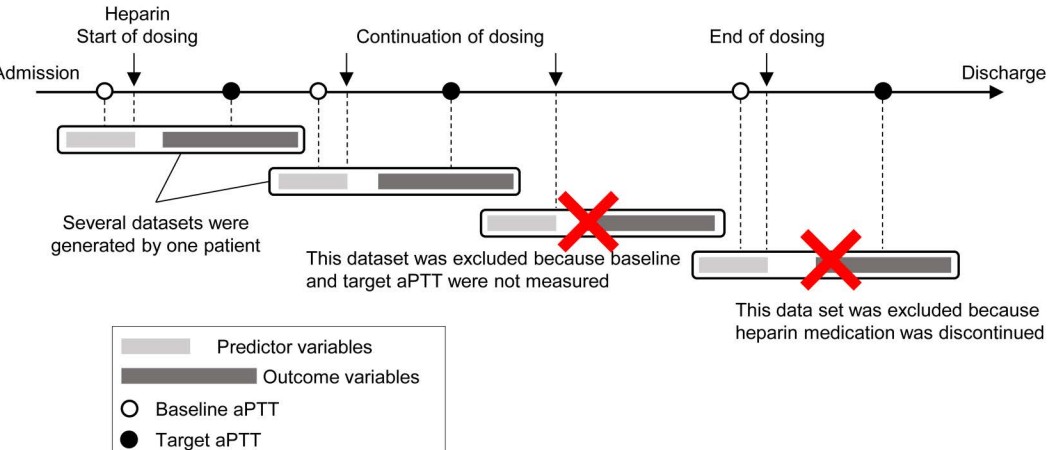

**Fig 1. Multiple dataset creation and exclusion criteria from each patient.** The dataset was created during entry and exit from the ICU/ECU while heparin was administered and dosing was continued. The dataset of interest included data in which the aPTT was measured prior to heparin administration and the target aPTT was measured between 6 and 24 h post-administration.

replacement therapy [CRRT], and Impella usage), coagulation markers (baseline activated partial thromboplastin time [aPTT], prothrombin time-international normalized ratio [PT-INR], fibrinogen, antithrombin III [AT-III], and D-dimer), biochemical markers (albumin, total bilirubin [T-Bil], aspartate aminotransferase [AST], alanine aminotransferase [ALT], lactate dehydrogenase [LDH], serum creatinine, and calcium), hematological parameters (white blood cell count, erythrocyte count, hemoglobin, hematocrit, mean corpuscular volume [MCV], mean corpuscular hemoglobin [MCH], mean corpuscular hemoglobin concentration [MCHC], and platelet count), and treatment-related metrics (urine output volume, total fluid volume [TFV], plasma product volume, platelet preparation dosage, erythrocyte product volume, total transfusion product volume [TPV], and blood heparin concentration before and after the start of administration [BHC before SA and BHC after SA]).

A previously reported correlation exists between aPTT and blood heparin concentration [14]. Therefore, we derived the BHC before SA and after SA as predictor variables. The BHC before SA was calculated by dividing the total heparin administered between the reference aPTT measurement and the start of heparin administration by the elapsed time, body weight, and CBV. CBV was estimated using height and weight. Similarly, the BHC after SA was calculated based on the total heparin administered from the start of administration to the time the target aPTT was measured. For bolus administration, the heparin dose at the time of administration was used, while for continuous administration, the total dose from start to end was considered.

For predicting the response to the first heparin dose, we used 45 variables as static data, excluding BHC before SA. The predictor variables were the most recent values recorded within 24 h before heparin administration in the ICU, except for balance-related variables (inputs, outputs) and medications. Urine output, classified as a balance output, was defined as the total daily volume recorded 24 h before heparin administration. Input variables represented the cumulative dose administered within 24 h preceding heparin administration. The BHC before and after SA calculations were performed as described earlier.

Outliers in static data were identified using the 3σ method, and both outliers and missing values were imputed using the five-nearest neighbor imputation method. Categorical variables, including gender, blood type, underlying disease, surgery status, ECMO treatment, and ECMO combined with other therapies, were processed using one-hot encoding. Numeric variables, such as blood pressure, were standardized, with standardization applied before imputation.

For cases involving multiple heparin administrations, variables were categorized into (i) static data and (ii) a combination of time-series and static data. Seven variables were treated as time-series data, including six vital signs and one balance output variable (urine volume). Time-series data were processed using interval spline interpolation and the median of five neighboring points. Data points were recorded hourly for 24 h before heparin administration, resulting in 168 time-series variables. Urine output was converted into hourly measurements. Alongside these 168 time-series variables, 39 static variables related to patient information, laboratory results, and medications were included, yielding 207 variables comprising both static and time-series data.

### Splitting and generating datasets

Datasets were generated for each heparin administration starting point (Fig 1). The dataset was split into training and test sets in an 8:2 ratio, with the training set used for model building and the test set used for accuracy assessment. Stratified five-fold cross-validation was performed. During cross-validation, the training and test sets did not include data from the same patient. Oversampling and undersampling techniques were applied only to the model with the best ROC AUC to address the imbalanced proportions of the three target variable classes. The purpose was to verify if resampling could further improve the accuracy of the model. Resampling was performed only on training data to confirm and evaluate the ability to classify even unbalanced test data. This resampling ensured that the ratio of sub-therapeutic to normal-therapeutic to supra-therapeutic classes was balanced at 1:1:1.

## Construction of prognostic model

A prognostic model was constructed using a multineural network, that is, a multilayer neural network, and support vector classification (SVC), random forest, extreme gradient boosting (XGB), light gradient boosting machine (LightGBM), and logistic regression. For the multineural network, we used recurrent neural network (RNN) when dealing with time series information in the prediction after multiple heparin doses. Hyperparameter tuning was performed using grid search only for the multineural network. Detailed conditions such as hyperparameters are summarized in S2 Table in S2 File. Data processing and manipulation were performed using Python (version 3.8.5).

## Performance evaluation

Macro averages were used to classify patients into the three target categories. An accuracy index was calculated for each class, and the overall average value was obtained, as detailed in S3 Table in S2 File. Model performance was assessed based on accuracy, macro receiver operating characteristic (ROC) area under the curve (AUC), macro precision–recall (PR) AUC, macro F1 score, macro precision, and macro recall. Prediction accuracy was evaluated using five different test datasets, with the final accuracy determined as the average of the five results.

To interpret the relationship between predictor variables and classification outcomes, we performed SHAP analysis using the SHAP package in Python. SHAP values quantify the impact of each feature by comparing the prediction accuracy of models built with and without the feature. Detailed calculation methods can be accessed in the package documentation (https://shap.readthedocs.io/en/latest/api.html). The analysis results were visualized using a beeswarm diagram, where red plots represent high feature values. A cluster of red dots on the right side of the diagram indicates that higher feature values increase the likelihood of classification into that category. The figure also displays the absolute average SHAP values. For the model with the highest ROC AUC, we identified and listed the top 10 variables with the greatest impact on predictions.

## Statistical analysis

The data are presented as n (%) or mean value±standard deviation. The Mann–Whitney U test and chi-square test were performed to compare variables in each dataset between the first and multiple heparin infusion group. The criterion for statistical significance was $p < 0.05$. Sample size calculations were performed ex post using the pROC package in R to evaluate ROC AUC. We had more than the required number of data, 95 for the first dose group and 69 for the multiple dose group, under ROC AUC of 0.7, power of 0.90, and one-tailed test, particularly as the supra-therapeutic group was approximately 20–30% of the other groups.

## Results

### Study participants

A total of 1,538 patients were admitted to the ICU and anticoagulated with heparin. Of these, 579 were excluded as they did not meet the eligibility criteria for the aPTT prediction task (Fig 2). Consequently, 837 and 665 patients were included in the model construction and evaluation as first-time and multiple-dose heparin cases, comprising 837 and 3,672 data points, respectively, from the start of heparin administration (Fig 2). Some cases, such as discontinued heparin administration, were included in the first-infusion model but not in the multiple-infusion model. Conversely, cases where the target aPTT was not measured were included only in the multiple-infusion model. This relationship is illustrated in S1 Fig in S1 File.

Table 1 summarizes the patient backgrounds used for constructing and evaluating the prediction models. Patient backgrounds at each facility are provided in S5 Table in S3 File, while those categorized by baseline aPTT and target aPTT classes are detailed in S6 and S7 Tables in S3 File, respectively.

Patients in the first-infusion group were significantly older than those in the multiple-infusion group, with a mean age of 70.42 years compared to 67.64 years (p < 0.001). The multiple-infusion group had a higher mean height (163.29 cm vs. 162.38 cm, p = 0.008) and weight (65.75 kg vs. 62.96 kg, p < 0.001). More patients in the multiple-infusion group received ECMO treatment (17.8% vs. 11.5%, p < 0.001) and CRRT (5.5% vs. 2.9%, p = 0.029). Baseline aPTT was significantly higher in the multiple-infusion group (55.08 vs. 48.51 s, p < 0.001), suggesting potential differences in coagulation status at the initiation of treatment.

Fig 3 illustrates the distribution of aPTT values following first and multiple heparin infusions. In the first-infusion group, the distribution of each therapeutic class was more imbalanced compared to the multiple-infusion group. Greater variability was observed in the first heparin infusion (Fig 3A) compared to multiple infusions (Fig 3B), likely reflecting individual clinicians' dosing decisions, which may have resulted in both over- and under-administration of heparin. The distinct spikes at 100 and 180 s are due to record censoring values set by the facility.

The classifications of the reference aPTT for the top 10 major injuries are summarized in S8 Table in S3 File, accounting for 70% of all cases. Among them, acute transmural myocardial infarction of the anterior wall was the most common.

## Prediction of aPTT for first heparin infusion

The performance evaluation of the aPTT prediction models for the first heparin infusion is presented in Table 2 and S9A Table in S3 File. The random forest model achieved the highest ROC AUC (0.707), indicating its moderate ability to distinguish between the three therapeutic aPTT classes (sub-therapeutic, normal-therapeutic, and supra-therapeutic). The model also achieved the highest PR AUC (0.539) and precision (0.585), demonstrating its effectiveness in predicting true positive outcomes. LightGBM performed reasonably well, with a ROC AUC of 0.682 and a PR AUC of 0.504—slightly lower than those of the random forest model but still competitive. The SVC and XGB models exhibited moderate performance, with ROC AUCs of 0.701 and 0.664, respectively. While the random forest model excelled in ROC AUC and precision, its F1 score (0.453) and recall (0.455) were comparatively lower, suggesting that it may have missed some

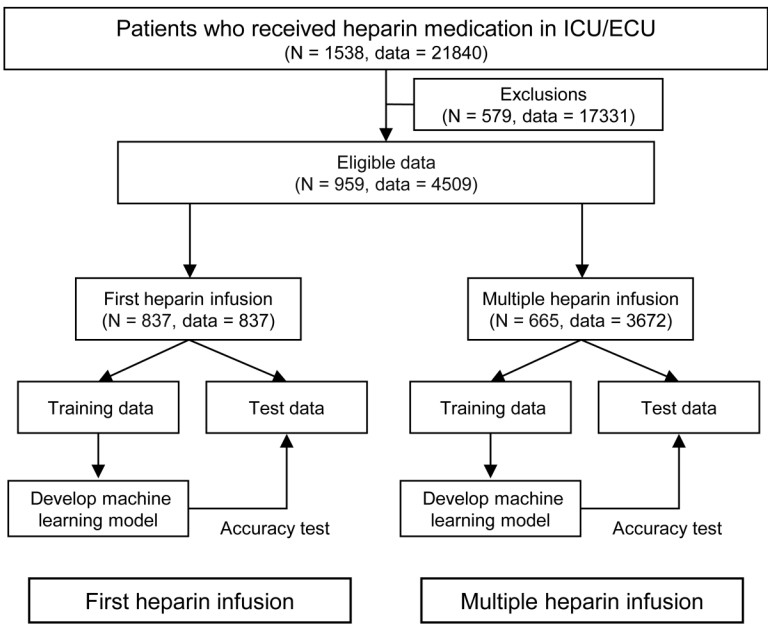

**Fig 2. Data processing and model construction procedure.**

**Table 1. Patient characteristics.**

| | First heparin infusion | Multiple heparin infusions | p-value | p<0.05 |
|---|---|---|---|---|
| | n=837 | n=665 | | |
| | 837 data | 3672 data | | |
| **Patient information** | | | | |
| Sex | | | 0.028 | * |
| Male (%) | 562 (67.1%) | 2607 (71.0%) | | |
| Female (%) | 275 (32.9%) | 1065 (29.0%) | | |
| Age (years) | 70.42±13.99 | 67.64±14.69 | p<0.001 | * |
| Height (cm) | 162.38±10.07 | 163.29±9.92 | 0.008 | * |
| Weight (kg) | 62.96±16.17 | 65.75±16.48 | p<0.001 | * |
| Blood types | | | 0.207 | |
| A | 309 (41.7%) | 1527 (44.0%) | | |
| B | 171 (23.1%) | 694 (20.0%) | | |
| O | 185 (25.0%) | 851 (24.5%) | | |
| AB | 76 (10.3%) | 402 (11.6%) | | |
| CBV (L) | 4.39±0.91 | 4.56±0.91 | p<0.001 | * |
| Disease | | | p<0.001 | * |
| Circulatory disease | 576 (68.8%) | 2234 (60.8%) | | |
| Cerebrovascular disease | 66 (7.9%) | 65 (1.8%) | | |
| Other | 195 (23.3%) | 1373 (37.4%) | | |
| Surgery | | | p<0.001 | * |
| No | 749 (89.5%) | 3581 (97.5%) | | |
| Yes | 88 (10.5%) | 91 (2.5%) | | |
| **ECMO treatment** | | | | |
| ECMO | | | p<0.001 | * |
| No | 741 (88.5%) | 3019 (82.2%) | | |
| Yes | 96 (11.5%) | 653 (17.8%) | | |
| **Combination treatment with ECMO** | | | | |
| CRRT | | | 0.029 | * |
| No | 813 (97.1%) | 3505 (95.5%) | | |
| Yes | 24 (2.9%) | 167 (5.5%) | | |
| Impella | | | 0.803 | |
| No | 828 (98.9%) | 3636 (99.0%) | | |
| Yes | 9 (1.1%) | 36 (1.0%) | | |
| aPTT | | | | |
| Baseline aPTT | 48.51±44.26 | 55.08±26.92 | p<0.001 | * |
| Elapsed time (h) | 16.68±5.82 | 15.93±6.42 | p<0.001 | * |
| **Laboratory data** | | | | |
| Albumin (g/dL) | 3.34±0.70 | 2.68±0.56 | p<0.001 | * |
| T-Bil (mg/dL) | 0.89±0.59 | 1.01±0.85 | 0.121 | |
| AST (U/L) | 105.99±166.48 | 98.11±170.33 | 0.023 | * |
| ALT (U/L) | 53.03±77.40 | 60.40±87.45 | 0.006 | * |
| LDH (U/L) | 444.05±339.78 | 490.38±358.47 | p<0.001 | * |
| Serum creatinine (mg/dL) | 1.27±0.85 | 1.24±0.84 | 0.005 | * |
| Calcium (mg/dL) | 8.53±0.72 | 8.02±0.63 | p<0.001 | * |
| White blood cell count (/μL) | 9985.77±3946.73 | 9708.06±3943.14 | 0.054 | |

*(Continued)*

**Table 1.** (Continued)

| | First heparin infusion | Multiple heparin infusions | p-value | p<0.05 |
|---|---|---|---|---|
| Erythrocyte count (10⁶/μL) | 4.09±0.77 | 3.7±0.66 | p<0.001 | * |
| Hemoglobin (g/dL) | 12.53±2.33 | 11.24±1.91 | p<0.001 | * |
| Hematocrit (%) | 37.16±6.63 | 33.47±5.51 | p<0.001 | * |
| MCV (fL) | 91.47±5.97 | 90.67±5.03 | p<0.001 | * |
| MCH (pg) | 30.84±1.95 | 30.44±1.67 | p<0.001 | * |
| MCHC (g/dL) | 33.68±1.28 | 33.56±1.23 | 0.001 | * |
| Platelet count (10³/μL) | 187.43±72.24 | 161.43±82.51 | p<0.001 | * |
| PT-INR | 1.23±0.42 | 1.25±0.25 | p<0.001 | * |
| Fibrinogen (mg/dL) | 356.13±131.34 | 389.73±145.99 | p<0.001 | * |
| AT-III (%) | 81.91±20.81 | 73.99±16.91 | p<0.001 | * |
| D-dimer (μg/mL) | 5.70±9.46 | 6.55±8.93 | p<0.001 | * |
| **Vital signs** | | | | |
| Body temperature (°C) | 36.58±0.98 | 36.80±0.88 | p<0.001 | * |
| Respiratory rate (/min) | 14.11±5.51 | 12.97±5.27 | 0.02 | * |
| Pulse rate (/min) | 83.80±20.37 | 81.11±17.95 | 0.05 | * |
| SpO2 (%) | 97.03±2.49 | 96.72±2.82 | 0.007 | * |
| SBP (mmHg) | 119.73±23.36 | 116.59±21.28 | 0.001 | * |
| DBP (mmHg) | 71.14±16.25 | 68.30±14.48 | p<0.001 | * |
| **Balance outputs** | | | | |
| Urine volume (mL/day) | 782.04±776.24 | 1638.40±1106.59 | p<0.001 | * |
| **Balance inputs** | | | | |
| TFV (mL) | 2490.95±1774.68 | 3017.37±1937.52 | p<0.001 | * |
| Plasma product volume (mL) | 61.94±303.88 | 58.89±290.00 | 0.117 | |
| Only for patients administered the above (mL) | 1036.80±737.65 | 766.81±744.26 | 0.001 | * |
| Platelet preparation dosage (mL) | 9.86±71.56 | 12.70±70.61 | 0.036 | * |
| Only for patients administered the above (mL) | 412.50±224.71 | 323.96±162.69 | 0.035 | * |
| Erythrocyte production volume (mL) | 94.34±367.86 | 113.58±337.86 | p<0.001 | * |
| Only for patients administered the above (mL) | 897.27±756.05 | 622.48±556.00 | p<0.001 | * |
| TPV (mL) | 166.13±697.44 | 185.17±634.89 | p<0.001 | * |
| Only for patients administered the above (mL) | 1479.26±1552.18 | 913.91±1150.94 | p<0.001 | * |
| **Medication (heparin dosing)** | | | | |
| BHC before SA (Unit/kg/h/L) | – | 1.48±1.41 | – | – |
| BHC after SA (Unit/kg/h/L) | 1.85±1.21 | 1.63±1.33 | p<0.001 | * |

CBV: circulation blood volume; ECMO: extracorporeal membrane oxygenation; CRRT: continuous renal replacement therapy; aPTT: activated partial thromboplastin time; Elapsed time: elapsed time from baseline aPTT measurement to target aPTT measurement; T-Bil: total bilirubin; AST: aspartate aminotransferase; ALT: alanine aminotransferase; LDH: lactate dehydrogenase; MCV: mean corpuscular volume; MCH: mean corpuscular hemoglobin; MCHC: mean corpuscular hemoglobin concentration; PT-INR: prothrombin time-international normalized ratio; AT-III: antithrombin III; SpO2: saturation of percutaneous oxygen; SBP: systolic blood pressure; DBP: diastolic blood pressure; Urine volume: amount of urine output; TFV: total fluid volume; TPV: total transfusion product volume; BHC before SA: blood heparin concentration before the starting point of administration; BHC after SA: blood heparin concentration after the starting point of administration.

Data are presented as n (%) or mean value±standard deviation.

cases, particularly in the sub-therapeutic and supra-therapeutic ranges. The contributions of these variables are illustrated in Fig 4A and S2A Fig in S1 File. Among the 45 variables treated with static data, the contributions of the administration of baseline aPTT and blood heparin concentration to the model were the highest.

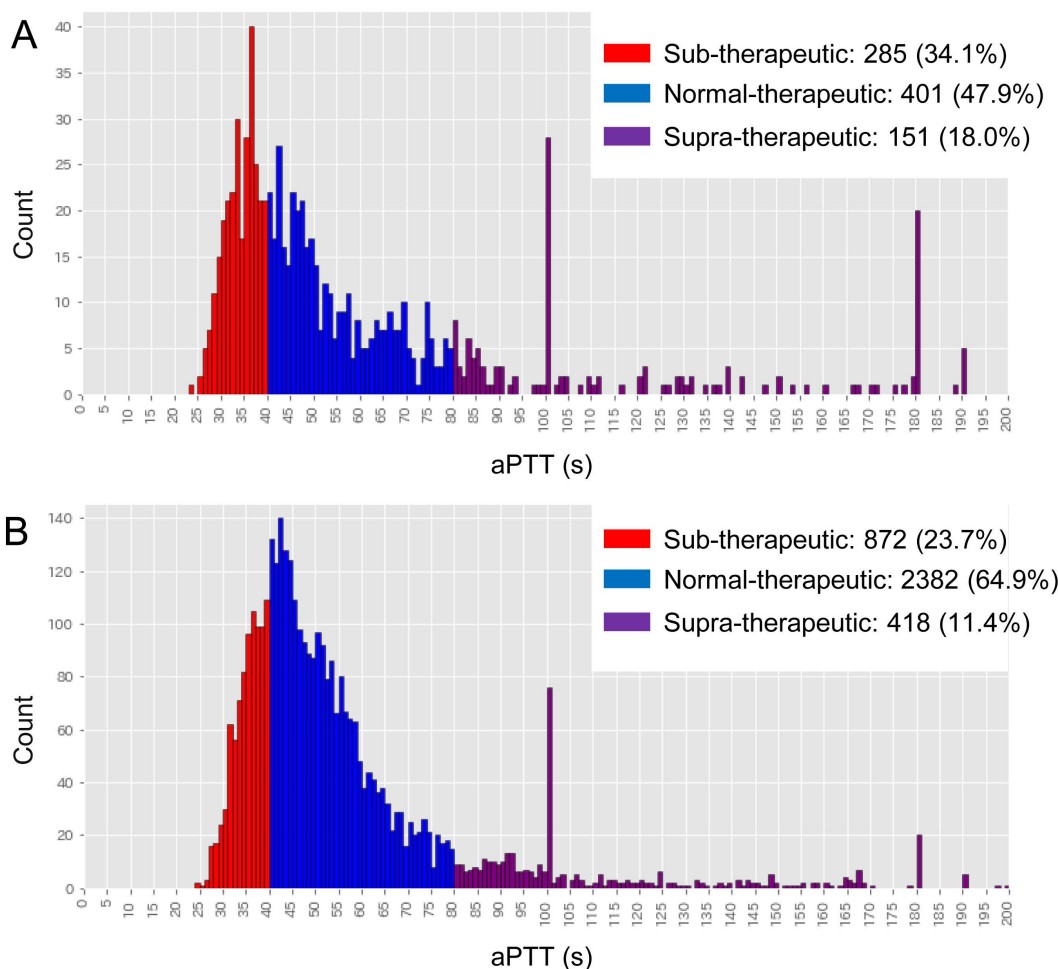

**Fig 3. Histogram of aPTT values after the (A) first dose and (B) multiple doses of heparin.**

**Table 2. Predictive performance of aPTT for first heparin infusion.**

| Data | Method | ROC-AUC | PR-AUC | $F_1$ | Precision | Recall | Accuracy |
|------|--------|---------|--------|-------|-----------|--------|----------|
| Static | Logistic regression | 0.680 ± 0.030 | 0.506 ±0.040 | 0.487 ± 0.060 | 0.512 ± 0.070 | 0.481 ± 0.048 | 0.533 ± 0.035 |
| | SVC | 0.701 ± 0.029 | 0.536 ± 0.038 | 0.474 ± 0.050 | 0.578 ± 0.061 | 0.469 ± 0.038 | 0.551 ± 0.031 |
| | XGB | 0.664 ± 0.025 | 0.483 ± 0.034 | 0.470 ± 0.056 | 0.492 ± 0.062 | 0.464 ± 0.051 | 0.520 ± 0.045 |
| | Random forest | 0.707 ± 0.030 | 0.539 ± 0.046 | 0.453 ± 0.036 | 0.585 ± 0.067 | 0.455 ± 0.029 | 0.550 ± 0.029 |
| | LightGBM | 0.682 ± 0.040 | 0.504 ± 0.052 | 0.490 ± 0.051 | 0.525 ± 0.037 | 0.486 ± 0.049 | 0.540 ± 0.029 |

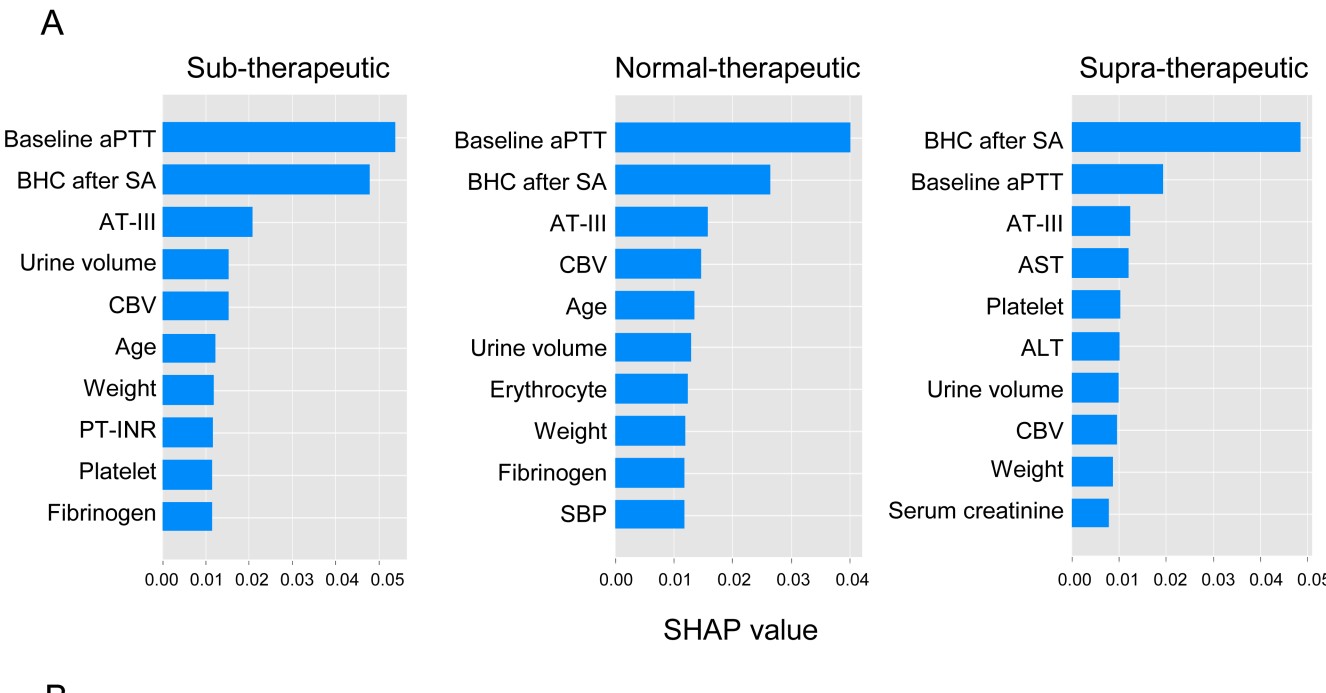

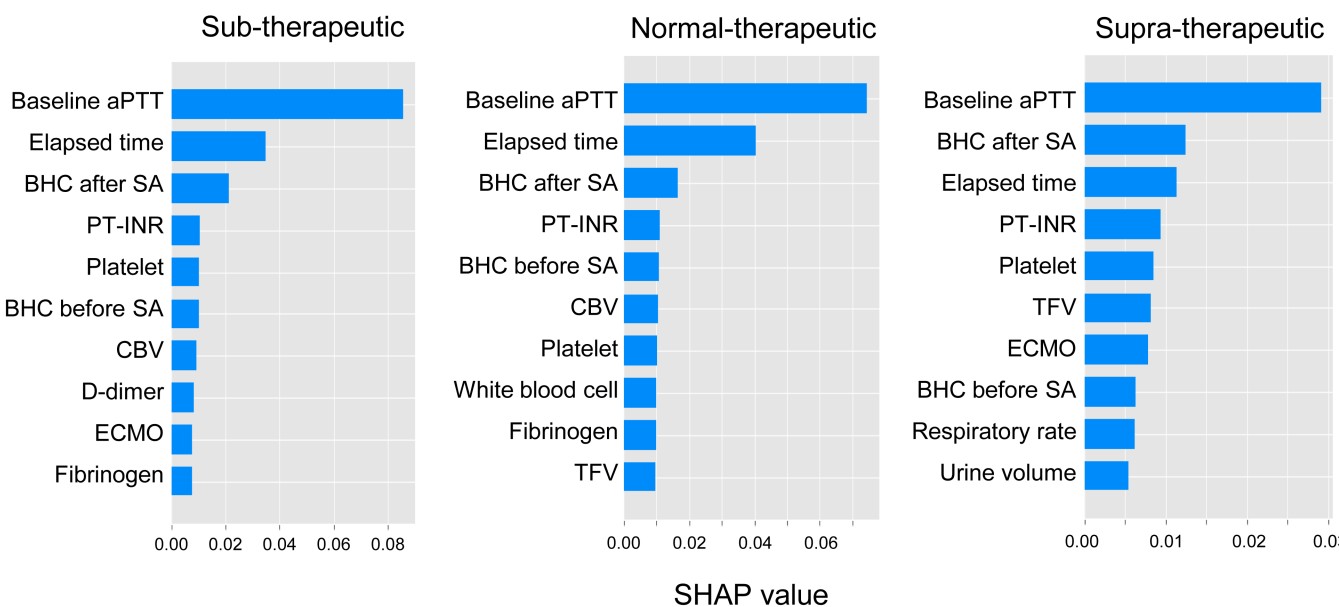

**Fig 4. Contributing variables for predictive performance in (A) first and (B) multiple heparin infusion patients.** The contributions to the prediction model were calculated as follows. For the first and multiple heparin infusion patient prediction, the random forest model with data imputed based on static data was used. aPTT: activated partial thromboplastin time; BHC after SA: blood heparin concentration after the starting point of administration; AT-III: antithrombin III; Urine volume: amount of urine output; CBV: circulation blood volume; PT-INR: prothrombin time-international normalized ratio; SBP: systolic blood pressure; AST: aspartate aminotransferase; ALT: alanine aminotransferase; Elapsed time: elapsed time from baseline aPTT measurement to target aPTT measurement; BHC before SA: blood heparin concentration before the starting point of administration; ECMO: extracorporeal membrane oxygenation; TFV: total fluid volume.

## Prediction of aPTT for multiple heparin infusion

The performance scores of the aPTT prediction models for multiple heparin infusions are presented in Table 3 and S9B Table in S3 File. The random forest model with static data achieved the highest performance, with a ROC AUC of 0.732 and PR AUC of 0.551, indicating its superior ability to classify aPTT values across multiple infusions accurately. Its precision (0.589) was also the highest, making it particularly effective in identifying true positive outcomes. Although models that incorporated time-series data (e.g., vital signs) were also evaluated, they offered only minimal improvements. For example, LightGBM with time-series data achieved a ROC AUC of 0.715, which was slightly lower than the 0.727 obtained using static data alone. The random forest model showed lower F1 scores (0.420) and recall (0.422), suggesting that while it was precise, it might have missed some positive cases, similar to its performance with first-dose predictions.

The contributions of these variables to the model are illustrated in Fig 4B and S2B Fig in S1 File. The contribution of the baseline aPTT to the model was the highest among the 46 variables treated with static data.

Oversampling and undersampling were performed to compare the data accuracy, with the model achieving high performance in the F1score and recall, respectively (S10 Table in S3 File).

## Discussion

This study utilized data from an extensive database of Japanese hospitals to develop an ML model for predicting the appropriate aPTT range in ICU patients receiving heparin. The random forest model achieved the highest accuracy in

**Table 3. Predictive performance of aPTT for multiple heparin infusions.**

| Data | Method | ROC-AUC | PR-AUC | $F_1$ | Precision | Recall | Accuracy |
|---|---|---|---|---|---|---|---|
| Static | Logistic regression | 0.689 ± 0.020 | 0.502 ± 0.030 | 0.417 ± 0.038 | 0.600 ± 0.040 | 0.415 ± 0.026 | 0.666 ± 0.024 |
| | SVC | 0.690 ± 0.023 | 0.496 ± 0.026 | 0.400 ± 0.044 | 0.559 ± 0.055 | 0.403 ± 0.029 | 0.655 ± 0.024 |
| | XGB | 0.721 ± 0.018 | 0.533 ± 0.025 | 0.474 ± 0.029 | 0.587 ± 0.039 | 0.460 ± 0.024 | 0.686 ± 0.022 |
| | Random forest | 0.732 ± 0.022 | 0.551 ± 0.030 | 0.420 ± 0.030 | 0.589 ± 0.054 | 0.422 ± 0.020 | 0.681 ± 0.025 |
| | LightGBM | 0.727 ± 0.015 | 0.543 ± 0.026 | 0.479 ± 0.020 | 0.611 ± 0.054 | 0.463 ± 0.015 | 0.688 ± 0.015 |
| | Multineural network | 0.685 ± 0.030 | 0.500 ± 0.038 | 0.358 ± 0.068 | 0.597 ± 0.058 | 0.382 ± 0.038 | 0.662 ± 0.036 |
| Time-series & static | Logistic regression | 0.669 ± 0.021 | 0.471 ± 0.031 | 0.416 ± 0.043 | 0.526 ± 0.046 | 0.412 ± 0.027 | 0.649 ± 0.025 |
| | SVC | 0.672 ± 0.020 | 0.476 ± 0.032 | 0.395 ± 0.036 | 0.546 ± 0.086 | 0.400 ± 0.023 | 0.655 ± 0.026 |
| | XGB | 0.708 ± 0.014 | 0.525 ± 0.031 | 0.463 ± 0.026 | 0.595 ± 0.062 | 0.450 ± 0.017 | 0.684 ± 0.009 |
| | Random forest | 0.691 ± 0.018 | 0.510 ± 0.030 | 0.390 ± 0.031 | 0.622 ± 0.106 | 0.398 ± 0.020 | 0.667 ± 0.023 |
| | LightGBM | 0.715 ± 0.020 | 0.531 ± 0.035 | 0.466 ± 0.033 | 0.594 ± 0.060 | 0.454 ± 0.024 | 0.687 ± 0.013 |
| | Multineural network | 0.698 ± 0.018 | 0.512 ± 0.034 | 0.384 ± 0.057 | 0.618 ± 0.059 | 0.396 ± 0.037 | 0.669 ± 0.028 |

SVC: support vector classification; XGB: extreme gradient boosting; LightGBM: light gradient boosting machine; ROC: receiver operating characteristic; AUC: area under the curve; PR: precision–recall. Data are presented as mean value ± standard deviation.

predicting heparin response, with ROC AUC values of 0.707 for the first infusion and 0.732 for multiple infusions. Although these models were effective in identifying true positive cases, their overall predictive performance remains limited, highlighting the need for further refinement. Notably, despite incorporating both static and dynamic features to enhance prediction accuracy, no significant improvement was observed when using dynamic features compared to static ones. To further validate the appropriateness of using complex models such as Random Forests, we trained a logistic regression model as a simple baseline. The logistic regression model, which assumes linear relationships among variables, consistently underperformed compared to the more complex models across all performance metrics in both first-dose and multiple-dose scenarios (Table 2 and S3 Table in S2 File and S9 Table in S3 File). This performance gap indicates that non-linear relationships likely exist within the data, thereby justifying the use of non-linear models for predicting aPTT responses.

Previous studies have demonstrated that key factors influencing the pharmacokinetics and pharmacodynamics of UFH—such as patient age, weight-adjusted bolus doses, maintenance doses, and aPTT values—must be considered in dosing models [15,16]. However, most studies have primarily focused on predicting and validating responses after the first dose. In contrast, this study is the first to validate an ML model for predicting aPTT values before and after heparin administration in ICU patients, incorporating both static data (e.g., age, blood test results) and time-series data (e.g., vital signs) specifically for aPTT prediction after multiple heparin infusions. Consistent with prior research, our findings indicate that baseline aPTT and blood heparin concentration significantly contribute to model performance, alongside heparin-related features [5]. However, contrary to expectations, incorporating time-series data did not improve prediction accuracy. A possible explanation is that the most influential predictive variables (Fig 4) were primarily laboratory test results, whereas vital sign variables modeled as time-series data had minimal contribution. These findings suggest that, even in ICU patients requiring devices such as ECMO, incorporating time-series data, including vital signs, may not substantially improve aPTT prediction accuracy. Furthermore, baseline aPTT and blood heparin concentration emerged as the strongest predictive factors, accounting for the majority of variability in aPTT responses. This suggests that the addition of further static variables is unlikely to provide significant improvements in model performance. Conversely, a previous study demonstrated that incorporating a time series of features, such as the therapeutic intervention scoring system 10 (TISS-10), simplified acute physiology score (SAPS-II), SOFA, and acute physiology and chronic health evaluation II (APACHE II), instead of only static and aggregated features, and utilizing a RNN achieved the highest performance in predicting aPTT after heparin treatment [6]. These findings suggest that incorporating time-series data of severity scores, rather than vital signs, into the model may be beneficial.

The distribution of predicted aPTT values differed between the first-dose and multiple-dose patients (Fig 3). While prediction accuracy varied across models, the random forest model consistently achieved higher ROC AUC and other accuracy metrics—except for the F1 score and recall—in the multiple-dose group compared with the first-dose group. Prior studies have shown that ML-based models improve in accuracy when trained on larger datasets, although imbalanced data can negatively affect classification performance [17]. Although the impact of data imbalance cannot be ignored, our findings suggest that developing a more accurate model for the multiple-dose group is feasible.

Furthermore, compared to previous studies, Boie et al. used an RNN to predict the effects of heparin administration 24 h post-treatment in a cohort of 5,742 patients [6]. They retrained and evaluated models from prior studies and found that their highest-performing model achieved an F1 score of 0.40 and an overall accuracy of 0.82. In contrast, our model achieved an F1 score of 0.490, indicating a comparable or superior level of accuracy.

This study highlights the potential of ML models in predicting heparin response, with several key strengths: (i) the use of a comprehensive database covering six hospitals in Japan, (ii) the integration of static and dynamic features for improved prediction, and (iii) the identification of crucial predictors for both initial and multiple dosing scenarios, offering valuable insights into factors affecting model accuracy. Integrating previous findings with our results suggests that developing a predictive model using random forest or RNNs—incorporating three key factors: static variables influencing aPTT (e.g., coagulation factors, inflammatory markers), time-series data related to severity scores (e.g., SOFA, APACHE), and precise information on cumulative heparin dosage—could enhance generalizability and clinical applicability.

However, several limitations must be considered. First, the relatively small dataset may have restricted model performance, as larger datasets generally improve robustness and predictive accuracy. Second, data replication was used for the multiple-dose model, treating heparin administration and aPTT measurements as independent events. Although common in medical studies with limited data, this approach may introduce bias by effectively duplicating cases [18]. Third, assumptions made to handle missing data and unclear reporting could have introduced bias and reduced generalizability. Additionally, the model's complexity, particularly given the dataset size, may lead to overfitting. Furthermore, this study did not account for variations in heparin dosing protocols across different hospitals, which may influence patient outcomes. Additionally, the elapsed time between baseline and target aPTT measurements was included as a variable, but it may introduce bias related to patient status. Future research should consider adjusting for inter-hospital variability to enhance external validity. Finally, while this retrospective study provided valuable insights, prospective validation is necessary to assess the model's applicability in real-time clinical decision-making. Addressing these limitations in future studies could improve model performance and ultimately contribute to better anticoagulation management in ICU patients.

## Conclusions

We utilized data from an extensive Japanese hospital database to develop an ML model for predicting the appropriate aPTT range in ICU patients receiving heparin. The random forest model achieved the highest accuracy, with ROC AUC values of 0.707 for the first dose and 0.732 for multiple doses. Although both static and dynamic features were incorporated to enhance prediction accuracy, no significant improvement was observed with dynamic features.

## Supporting information

**S1 File.** S1 Fig: A Venn diagram illustrating the number of data points used to construct and evaluate each prediction model. S2 Fig: Beeswarm diagrams showing contributing variables to predictive performance in both first and multiple heparin infusion models.
(DOCX)

**S2 File.** S1 Table: Exclusions in the data processing and model construction procedure. S2 Table: Hyperparameter tuning settings for each machine learning model used in the study. S3 Table: Confusion matrix and calculation methods for evaluation metrics.
(DOCX)

**S3 File.** S4 Table: Patient characteristics in the study population for first heparin infusion (A) and multiple infusions (B) at each facility. S5 Table: Patient characteristics across three classes of baseline aPTT for first heparin infusion (A) and multiple infusions (B). S6 Table: Patient characteristics across three classes of target aPTT for first heparin infusion (A) and multiple infusions (B). S7 Table: Predictive performance of aPTT in each class for first heparin infusion. S8 Table: Predictive performance of aPTT in each class for first heparin infusion (A) and multiple infusions (B). S9 Table: Predictive performance of aPTT for multiple heparin infusions.
(XLSX)

## Acknowledgments

The authors thank six hospitals for their contributions to the Tokushukai Medical Database.

## Author contributions

**Conceptualization:** Tadashi Kamio.

**Data curation:** Masaru Ikegami, Megumi Mizuno, Seiichiro Ishii, Hayato Tajima, Yoshihito Machida.

**Formal analysis:** Megumi Mizuno, Seiichiro Ishii, Hayato Tajima, Yoshihito Machida.

**Investigation:** Tadashi Kamio.

**Methodology:** Tadashi Kamio, Masaru Ikegami, Yoshihito Machida.

**Writing – original draft:** Tadashi Kamio.

**Writing – review & editing:** Masaru Ikegami, Kiyomitsu Fukaguchi.

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
