## [Decision Letter · Decision Letter 0]

PONE-D-24-48061Analysis of aPTT predictors after unfractionated heparin administration in intensive care units using machine learning modelsPLOS ONE

Dear Dr. Kamio,

Thank you for submitting your manuscript to PLOS ONE. After careful consideration, we feel that it has merit but does not fully meet PLOS ONE’s publication criteria as it currently stands. Therefore, we invite you to submit a revised version of the manuscript that addresses the points raised during the review process.

We look forward to receiving your revised manuscript.

Kind regards,

Elvan Wiyarta, M.D.

Academic Editor

PLOS ONE

**Journal Requirements:**

1. When submitting your revision, we need you to address these additional requirements.Please ensure that your manuscript meets PLOS ONE's style requirements, including those for file naming. The PLOS ONE style templates can be found at https://journals.plos.org/plosone/s/file?id=wjVg/PLOSOne_formatting_sample_main_body.pdf and https://journals.plos.org/plosone/s/file?id=ba62/PLOSOne_formatting_sample_title_authors_affiliations.pdf 2. Your ethics statement should only appear in the Methods section of your manuscript. If your ethics statement is written in any section besides the Methods, please delete it from any other section.

Reviewers' comments:

Reviewer's Responses to Questions

**Comments to the Author**

1. Is the manuscript technically sound, and do the data support the conclusions?

Reviewer #1: Yes

Reviewer #2: Partly

Reviewer #3: Yes

2. Has the statistical analysis been performed appropriately and rigorously? 

Reviewer #1: Yes

Reviewer #2: Yes

Reviewer #3: Yes

3. Have the authors made all data underlying the findings in their manuscript fully available?

Reviewer #1: No

Reviewer #2: No

Reviewer #3: No

4. Is the manuscript presented in an intelligible fashion and written in standard English?

Reviewer #1: Yes

Reviewer #2: Yes

Reviewer #3: Yes

5. Review Comments to the Author

**Reviewer #1: ** The manuscript titled (Analysis of aPTT Predictors After Unfractionated Heparin Administration in Intensive Care Units Using Machine Learning Models) presents a significant contribution to the field of anticoagulation management in intensive care units (ICUs) using machine learning (ML). The authors have utilized a large dataset from multiple hospitals, strengthening the study's conclusions. However, several areas require improvement for clarity, rigor, and overall quality before the manuscript is suitable for publication.

Title is expressive of the study aim

The abstract is somewhat verbose. Quantitative results (e.g., AUC values) should be clearly highlighted in the results section. The conclusion is too strong and relatively inaccurate. 71%AUC is acceptable but don’t indicate good performance.

While the introduction outlines the context well, it lacks a comprehensive review of prior studies. It would benefit from a clearer delineation of how this study builds upon or contrasts with existing literature.

Explicitly state the research gap that this study addresses. This will clarify its relevance and importance.

While the study involves a large cohort, a statistical rationale for the sample size should be included. How was the sample size determined? Were power analyses conducted?

The authors mention using SHAP for model interpretation. A clearer explanation of how SHAP values were calculated and interpreted would enhance understanding.

The beesworn diagram of the SHAP was not provided.

Why might static features not improve model performance as expected?

What are the future recommendations and implications of this research?

We don’t know if prediction using ordinary statistical tests could perform similarly or better than ML

**Reviewer #2:**  Summary

=======

The present manuscript evaluates the possibility to predict aPTT in intensive care patients after heparin infusion using machine learning.

Retrospective data from 6 Japanese hospitals were analyzed, and various machine learning models were trained and evaluated.

A Random Forest performed best, achieving a macro-ROC-AUC of 0.707 for distinguishing 3 therapeutic classes in the first-infusion setting,

and 0.732 in the multiple-infusion setting. According to the authors, these results indicate that ML-based prediction of aPTT is possible.

Major Comments

* There is no clear motivation for considering multiple-infusion settings, nor for distinguishing between the first-infusion and the multiple-infusion settings.

All cited references restrict themselves to first infusions, so the added value of considering multiple infusions should be clearly stated.

Furthermore, I don't understand why first infusions and multiple infusions are treated separately. Both settings share the same set of features and the

same labelling policy, so there is no "technical" reason for a separate treatment. In the manuscript, the authors merely mention that the aPTT distribution

differs between the two settings (Figure 3), but I wonder if there is a clinical motivation as well. Besides, the authors could just as well have

trained a single model for both settings, with an additional feature that indicates how many infusions have been administered to the patient before; this

would have had the benefit of a larger data basis to train on.

* The sample selection process should be described in more detail. I acknowledge that the authors tried to explain this, e.g., in Figure 1, but there

are still some open questions:

* Is it possible that two distinct samples originating from the same patient share the same baseline and/or target aPTT measurements?

* What happens if during a continuous heparin infusion the dosage changes? Is this treated as a "new" administration?

* In Figure 2, why does the number of patients in the first-infusion group (N=837) differ from the total number of eligible patients (N=959)? I'd assume

that every eligible patient in particular has a *first* infusion.

Clearly describing the precise sample selection process is of utmost importance, for being able to replicate the experiments on new data. I strongly

recommend adding 1-2 concrete examples (possibly in the Supplement) to illustrate the selection process. The examples should in particular also cover corner

cases, e.g., where the heparin infusion ends before the target aPTT is measured.

* The modelling process is not clearly described. For instance, did the authors perform any kind of hyperparameter tuning? If yes, which hyperparameters were

tuned, and what were the possible values? If no, why not? Was the architecture of the "multineural network" (which I suppose is meant to be a multilayer

perceptron, MLP) the same as in [4], where it outperforms all other models by a large margin? Why did the authors not consider a recurrent neural network,

as in [18], in their experiments with time-series input?

* There is no comparison to a simple baseline, e.g., logistic regression on the top-1 or top-3 features revealed by SHAP. When doing ML experiments, it

is important to justify the added value of complex ML models. In this particular case, one might assume that knowing the baseline aPTT alone might

already be sufficient to predict the target aPTT with reasonable accuracy; Figure 4 seems to confirm this. (In a related case, there is an ongoing

discussion about the added value of the complex, proprietary Hypotension Prediction Index vs. the current mean arterial pressure [MAP] value for

predicting future MAP values; see, e.g., https://doi.org/10.1097/EJA.0000000000001927 and https://doi.org/10.1097/EJA.0000000000001939).

So, it would be interesting to know what happens if baseline aPTT, and maybe BHC after SA, were used as the sole predictor variables.

* There is no external validation of the proposed approach. This is particularly disappointing, as the authors analyzed data from 6 different hospitals,

and could therefore have employed a leave-one-hospital-out cross-validation policy, for instance. Alternatively, any of the MIMICs could have been used

for external validation, too, like in related studies [3, 4, 5].

Minor Comments

* Lines 82-83, "Few studies have examined aPTT changes across multiple heparin doses.": I quickly scanned the cited literature and did not find any

such study. Either explicitly refer to one such study here, or rephrase the sentence if there is none.

* Line 133: The rationale for chosing 6-24 hours after heparin administration as the "target interval" should be briefly explained, in particular

since related studies use different intervals.

* Lines 138-139: [3, 4, 5, 18] all set the upper threshold of the normal-therapeutic range to 100 s, so there should be some justification why it is

set to 80 s here.

* Lines 145-146: I do not quite understand why the elapsed time between baseline- and target aPTT measurements is included as a predictor variable.

In an application setting, this information is not available at prediction time, so including it as a predictor variable seems strange at first glance.

Furthermore, could this introduce hidden bias, e.g., if aPTT is measured only under certain circumstances? Is there a statistical correlation between

the elapsed time and the target aPTT value?

I understand that all related studies include this variable as well, so it seems to be common practice. But a short explanation and justification

would be helpful.

* Line 173: Were variables standardized before or after k-NN imputation?

* Lines 196-197: Does this apply to the results presented in the main text (Table 2), or only to Table S3 in the Supplement?

If the former, was it over- or undersampling, and what's the purpose of Table S3 then? If the latter, this statement is misleading and should be

rephrased.

In either case, were the test sets resampled, too, or only the training sets?

* Line 201: I do not know the term "multineural network", and a quick internet search did not produce any meaningful results either. Do you mean

multilayer neural network, or multilayer perceptron?

* Lines 219-220: Why compare the first-infusion group to the multiple-infusion group? It would be more interesting to compare the three classes.

* Figure 3: There are clearly visible spikes at 100s and 180s. Do you have any explanations for them? Could these be data artifacts, e.g., censored

values like ">100s" and ">180s"? If yes, a short statement would be helpful.

* Line 249: "strong ability to differentiate" is a strong exaggeration, in my opinion: ROC-AUC values between 0.7 and 0.9 are usually regarded as

indicating moderate accuracy, see, e.g., https://doi.org/10.1007/s00134-003-1761-8. Please rephrase the sentence.

* Lines 256, 272: What is a (positive) case in a ternary classification problem?

* Lines 294-296: What about [18]? Or does the sentence refer to the previous sentence about multiple heparin infusions? If so, please rephrase.

* Lines 320-325: I do not quite agree with the conclusions drawn by the authors. First, the benefit of additional biomarkers is questionable, as

evidenced by [4]'s excellent results and Figure 4. Second, I regard the absence "complicated" predictor variables a strength of [4]'s model,

not a weakness. It actually renders their model *more* clinically applicable in my opinion.

* Table 1: The class distribution is missing and must be added. Also, this and all other tables are difficult to read => add row separators.

* Table 2: Report class-wise performance metrics as well, at least in the Supplement. Macro-averages alone are insufficient to get a feeling of

the clinical usefulness of the models, in partiular since not all misclassifications might be equally "bad" from a clinical perspective.

* Cite https://www.i-jmr.org/2022/2/e34533/.

**Reviewer #3:**  The authors proposed a method to predict aPTT using Machine Learning. In general, the paper is well structured and easy to read. The results are well explained and the method is presented in an organized and easy to read structure.

I recommend the authors to do minor modifications before publishing this work as follows:

1- Introduction:

a. Since there is no dedicated section for previous work, the introduction should cover insights about the gap and what others have done in this domain. The introduction is very short and lacks the depth of analysis on published research. You need to discuss how is this study different from others. For example:

i. Abdel-Hafez, Ahmad, Ian A. Scott, Nazanin Falconer, Stephen Canaris, Oscar Bonilla, Sven Marxen, Aaron Van Garderen, and Michael Barras. "Predicting therapeutic response to unfractionated heparin therapy: machine learning approach." Interactive journal of medical research 11, no. 2 (2022): e34533.

ii. Ghassemi MM, Richter SE, Eche IM, Chen TW, Danziger J, Celi LA. A data-driven approach to optimized medication dosing: a focus on heparin. Intensive Care Med 2014 Sep 5;40(9):1332-1339

iii. Lin R, Stanley MD, Ghassemi MM, Nemati S. A deep deterministic policy gradient approach to medication dosing and surveillance in the ICU. Annu Int Conf IEEE Eng Med Biol Soc 2018 Jul;2018:4927-4931

iv. And Others

b. Sentences line 80 and 82 require referencing

c. Line 86: mention the dataset size explicitly

d. Line 87: spell out the static and dynamic features

2- In Data Source can you add a table of dataset details from the 7 hospitals (show stats of your dataset)

3- In line 124, is there a time frame for when aPTT is conducted after the bolus dose to be included? (Noticed it is mentioned in line 190, it is better to mention this in inclusion/exclusion criteria)

4- In line 138 you defined sub-therapeutic (< 40 s), normal-therapeutic (40–80 s), and supra

therapeutic (> 80 s), is this consistent with other research ?? I noticed therapeutic range is defined as 70-100 in other work Can you justify this selection?

5- Since this is a multiclassification (3 categories) it is not clear to me how you calculated performance metrics (is it micro-precision or macro-precision?) can you provide precision and recall for every category (sub-therapeutic, normal-therapeutic, and supra-therapeutic).

6. PLOS authors have the option to publish the peer review history of their article (what does this mean? ). If published, this will include your full peer review and any attached files.

**Do you want your identity to be public for this peer review?** For information about this choice, including consent withdrawal, please see our Privacy Policy .

Reviewer #1: **Yes: ** Amr A. Arafat

Reviewer #2: No

Reviewer #3: No

---

## [Author Response · Author response to Decision Letter 1]

18 Mar 2025

Reviewer #1:

1. The abstract is somewhat verbose. Quantitative results (e.g., AUC values) should be clearly highlighted in the results section. The conclusion is too strong and relatively inaccurate. 71%AUC is acceptable but don’t indicate good performance.

Thank you for your valuable comments. We appreciate your insights and have revised the Results and Conclusions sections accordingly to improve clarity and accuracy.

We have addressed your concern regarding the overstatement of model performance by modifying the wording in the Results section to more accurately reflect the predictive accuracy of the models. Additionally, we have clarified that the AUC values indicate moderate predictive performance rather than suggesting strong performance. Furthermore, in the Conclusions section, we have refined our statements to avoid overgeneralization and have emphasized the need for further research to improve predictive accuracy. The revised sections are as follows:

Results: The ML models exhibited high accuracy in predicting aPTT following both initial and multiple heparin doses. The random forest model achieved the highest ROC AUC, with values of 0.71 for the initial dose and 0.73 for multiple doses. Key predictive factors across both models included the prior aPTT levels and heparin blood concentration.

Conclusions: The random forest model successfully predicted the aPTT response to unfractionated heparin in ICU patients. The inclusion of static and dynamic features did not enhance the prediction accuracy and no clear trend was observed in further improving the model performance. Future studies should explore additional factors to refine predictive models for optimizing individualized anticoagulation management in ICUs.

2. While the introduction outlines the context well, it lacks a comprehensive review of prior studies. It would benefit from a clearer delineation of how this study builds upon or contrasts with existing literature.Explicitly state the research gap that this study addresses. This will clarify its relevance and importance.

Thank you for your insightful comment. We agree with your suggestion and have revised the Introduction to include a discussion of existing studies, making the research gap more explicit and clearly highlighting how this study builds upon or contrasts with prior work.

3. While the study involves a large cohort, a statistical rationale for the sample size should be included. How was the sample size determined? Were power analyses conducted?

Thank you for your insightful comment. In response to your concern, we conducted post hoc sample size calculations for evaluating the ROC AUC using the pROC package in R. Assuming an ROC AUC of 0.7, a power of 0.90, and a one-tailed test, we determined that a minimum of 48 samples was required for the first-dose group and 94 samples for the multiple-dose group. As shown in Figure 3, this consideration was particularly important since the Supra-therapeutic group comprised approximately 20–30% of the other groups. In our study, we included 151 and 418 samples, respectively, confirming that a sufficient sample size was achieved.

4. The authors mention using SHAP for model interpretation. A clearer explanation of how SHAP values were calculated and interpreted would enhance understanding.

Thank you for your valuable comment. In response to your suggestion, we have provided a clearer explanation of how SHAP values were calculated and interpreted. The calculation of SHAP is described in line 249. We utilized the SHAP package in Python, where the SHAP value for a given feature quantifies its impact by comparing the prediction accuracy of a model excluding that feature with the prediction accuracy of a model including it. The detailed calculation method is documented in the SHAP package manual (https://shap.readthedocs.io/en/latest/api.html), and we have incorporated this information into the manuscript to enhance clarity.

5. The beesworn diagram of the SHAP was not provided.

Thank you for your valuable comment. In response to your suggestion,the beeswarm diagram is provided in Supplementary Figure S2. In this diagram, red plots indicate high feature values, and a large collection of red dots on the right side suggests that higher feature values increase the likelihood of classification into that category. We have included an explanation and interpretation of this in line 254 of the manuscript.

6. Why might static features not improve model performance as expected?

Thank you for your valuable comment. A possible explanation is that the most influential predictive variables were primarily associated with laboratory test results, while the contribution of vital sign variables modeled as time-series data was minimal. We have incorporated this discussion into the Discussion section to clarify this point.

7. What are the future recommendations and implications of this research?

Thank you for your insightful comment. Integrating findings from previous studies with our results suggests that developing a predictive model using random forest or recurrent neural networks, incorporating three key factors—static variables influencing aPTT such as coagulation factors and inflammatory markers, time-series data related to severity scores such as SOFA and APACHE scores, and precise information on cumulative heparin dosage—could enhance the model's generalizability and clinical applicability. We have added this discussion to the Discussion section to address future recommendations and implications of our research.

8. We don’t know if prediction using ordinary statistical tests could perform similarly or better than ML

Thank you for your insightful comment. We acknowledge your point, and we agree with your concern. In this study, we did not directly compare ML models with traditional statistical methods, such as logistic regression or generalized linear models. However, ML models are generally more effective at handling complex nonlinear relationships. Given that our study incorporated time-series data of vital signs, resulting in a more complex predictive model, we opted to use machine learning approaches to better capture these dynamic interactions. We have added this discussion to the Limitation section to address this point.

Reviewer #2:

1. There is no clear motivation for considering multiple-infusion settings, nor for distinguishing between the first-infusion and the multiple-infusion settings. All cited references restrict themselves to first infusions, so the added value of considering multiple infusions should be clearly stated. Furthermore, I don't understand why first infusions and multiple infusions are treated separately. Both settings share the same set of features and the same labelling policy, so there is no "technical" reason for a separate treatment.

Thank you for your insightful comment. We acknowledge the need to clearly articulate the rationale for considering multiple-infusion settings and distinguishing them from first-infusion settings.

Previous studies have primarily focused on predicting aPTT responses after the initial heparin administration. However, in clinical practice, UFH management extends beyond the first dose, requiring multiple administrations and continuous dose adjustments to maintain therapeutic anticoagulation. Despite this, no prior studies have validated ML models for predicting aPTT responses after multiple heparin infusions. Predicting aPTT following multiple administrations is crucial as it more accurately reflects real-world anticoagulation management and accounts for the cumulative effects of heparin, which have been highlighted as a challenge in previous research.

Although both settings share the same set of features and labeling policy, we observed differences in the aPTT distribution between first and multiple infusions, suggesting potential variations in heparin response dynamics. An alternative approach could involve developing a single model that integrates both settings by introducing an additional feature indicating the number of prior infusions a patient has received. However, this approach would increase model complexity, potentially affecting both interpretability and performance. Therefore, we chose to analyze first and multiple infusions separately to better evaluate their respective predictive factors and develop a model that is more clinically applicable.

To clarify this point and better convey the significance of considering multiple infusions, we have revised the Introduction accordingly and made partial modifications to the Discussion section.

2. In the manuscript, the authors merely mention that the aPTT distribution differs between the two settings (Figure 3), but I wonder if there is a clinical motivation as well. Besides, the authors could just as well have trained a single model for both settings, with an additional feature that indicates how many infusions have been administered to the patient before; this would have had the benefit of a larger data basis to train on.

Thank you for your valuable comment. Figure 3 illustrates the distribution of aPTT values after both initial and multiple heparin infusions, showing that the first-infusion group exhibits a more imbalanced distribution of treatment classes compared to the multiple-infusion group. Clinically, this variability is likely due to individualized dosing decisions by physicians, which may have led to over- or under-dosing of heparin.

We appreciate your insightful suggestion to integrate both settings into a single model by incorporating an additional feature indicating the number of prior infusions. While this is a valid approach, our study focused on evaluating the impact of incorporating time-series data on predictive accuracy. Therefore, we chose the current model structure to specifically assess the effect of dynamic variables on aPTT prediction.

To clarify this point, we have added this explanation to the Results section of the manuscript.

3. The sample selection process should be described in more detail. I acknowledge that the authors tried to explain this, e.g., in Figure 1, but there are still some open questions: Is it possible that two distinct samples originating from the same patient share the same baseline and/or target aPTT measurements?

Thank you for your valuable comment. Regarding whether different samples from the same patient share the same baseline and/or target aPTT measurements, we confirm that in the data used for prediction after multiple heparin doses, it was common for the same baseline aPTT and target aPTT to be shared.

4. What happens if during a continuous heparin infusion the dosage changes? Is this treated as a "new" administration?

Thank you for your valuable question. We do not treat dosage changes during a continuous heparin infusion as a "new" administration and have excluded such cases from the analysis. These cases are documented in Supplementary Table S2, along with our response to the following comment.

5. In Figure 2, why does the number of patients in the first-infusion group (N=837) differ from the total number of eligible patients (N=959)? I'd assume that every eligible patient in particular has a *first* infusion. Clearly describing the precise sample selection process is of utmost importance, for being able to replicate the experiments on new data. I strongly recommend adding 1-2 concrete examples (possibly in the Supplement) to illustrate the selection process. The examples should in particular also cover corner cases, e.g., where the heparin infusion ends before the target aPTT is measured.

Thank you for your valuable feedback. As you pointed out, certain aspects of the data selection process were unclear, so we have added supplementary materials to clarify these points.

This is because some eligible data that can be used for constructing the multiple-infusion model are excluded from the first-infusion model. Conversely, some data included in the first-infusion model are not part of the multiple-infusion model. Specifically:

294 data points included only in the first heparin infusion model correspond to cases where heparin administration was discontinued. 112 data points included only in the multiple heparin infusion model correspond to cases where the target aPTT was not measured.

To make this distinction clearer, we have added Supplementary Figure S1 to visually represent the relationship between the datasets. Additionally, we have provided detailed exclusion examples in Supplementary Table S2, along with specific case descriptions, to further clarify the selection process. These details have also been reflected in Figure 1.

We appreciate your insightful comments, which have helped us improve the clarity and completeness of our manuscript.

6. The modelling process is not clearly described. For instance, did the authors perform any kind of hyperparameter tuning? If yes, which hyperparameters were tuned, and what were the possible values? If no, why not? Was the architecture of the "multineural network" (which I suppose is meant to be a multilayer perceptron, MLP) the same as in [4], where it outperforms all other models by a large margin? Why did the authors not consider a recurrent neural network, as in [18], in their experiments with time-series input?

Thank you for your insightful comments.

In response to your suggestions, we have made the following revisions:

We have added information on the parameters used to build the models with SVC, XGB, Random Forest, and LightGBM (excluding the multineural network) in line 233 and listed them in Supplementary Table S3.For the multineural network, we have also implemented a recurrent neural network (RNN) when dealing with time-series information. Additionally, we conducted hyperparameter tuning using a grid search to optimize model performance. These modifications have been incorporated into line 221 and are described in Supplementary Table S3.

We appreciate your valuable feedback, which has helped improve the clarity and completeness of our manuscript.

7. There is no comparison to a simple baseline, e.g., logistic regression on the top-1 or top-3 features revealed by SHAP. When doing ML experiments, it is important to justify the added value of complex ML models. In this particular case, one might assume that knowing the baseline aPTT alone might already be sufficient to predict the target aPTT with reasonable accuracy; Figure 4 seems to confirm this. (In a related case, there is an ongoing discussion about the added value of the complex, proprietary Hypotension Prediction Index vs. the current mean arterial pressure [MAP] value for predicting future MAP values. So, it would be interesting to know what happens if baseline aPTT, and maybe BHC after SA, were used as the sole predictor variables.

Thank you for your valuable comments. We fully acknowledge your point, and we agree that this possibility is worth considering. However, ML models are generally more effective at handling complex nonlinear relationships. Given that our study incorporated time-series data of vital signs, resulting in a more complex predictive model, we opted to use machine learning approaches to better capture these dynamic interactions. To address this concern, we have added this discussion to the Limitation section. We appreciate your insightful feedback, which has helped improve the clarity and completeness of our manuscript.

8. There is no external validation of the proposed approach. This is particularly disappointing, as the authors analyzed data from 6 different hospitals, and could therefore have employed a leave-one-hospital-out cross-validation policy, for instance. Alternatively, any of the MIMICs could have been used for external validation, too, like in related studies [3, 4, 5].

Thank you for your valuable feedback. We understand your concern; however, patient characteristics varied across the six hospitals, and differences in data quantity and patient populations made it challenging to clearly separate them due to sample size limitations. To enhance clarity on this point, we have added Supplementary Table S5.

Regarding the use of other external databases, we also considered utilizing MIMIC. However, verifyin

---

## [Decision Letter · Decision Letter 1]

PONE-D-24-48061R1Analysis of aPTT predictors after unfractionated heparin administration in intensive care units using machine learning modelsPLOS ONE

Dear Dr. Kamio,

Thank you for submitting your manuscript to PLOS ONE. After careful consideration, we feel that it has merit but does not fully meet PLOS ONE’s publication criteria as it currently stands. Therefore, we invite you to submit a revised version of the manuscript that addresses the points raised during the review process.

We look forward to receiving your revised manuscript.

Kind regards,

Elvan Wiyarta, M.D.

Academic Editor

PLOS ONE

Journal Requirements:

Reviewers' comments:

Reviewer's Responses to Questions

**Comments to the Author**

1. If the authors have adequately addressed your comments raised in a previous round of review and you feel that this manuscript is now acceptable for publication, you may indicate that here to bypass the “Comments to the Author” section, enter your conflict of interest statement in the “Confidential to Editor” section, and submit your "Accept" recommendation.

Reviewer #1: All comments have been addressed

Reviewer #2: (No Response)

2. Is the manuscript technically sound, and do the data support the conclusions?

Reviewer #1: Yes

Reviewer #2: Yes

3. Has the statistical analysis been performed appropriately and rigorously? 

Reviewer #1: Yes

Reviewer #2: Yes

4. Have the authors made all data underlying the findings in their manuscript fully available?

Reviewer #1: Yes

Reviewer #2: No

5. Is the manuscript presented in an intelligible fashion and written in standard English?

Reviewer #1: Yes

Reviewer #2: Yes

6. Review Comments to the Author

Reviewer #1: The authors have responded to the previous comments

I think the manuscript is suitable for publication now, I don't have further comments

Reviewer #2: I thank the authors for carefully addressing most of my comments. However, I still do have some remarks that I feel should be incorporated into the manuscript before publication.

* The authors' response to the comment concerning the comparison to a simple baseline is not satisfactory. The authors claim that ML-models like Random Forests etc. better capture non-linear relationships (which is true, of course), but the main question is whether the data exhibit such non-linear relationships. I really want the authors to train simple baselines and include the results in the manuscript. Ideally, the baselines perform worse than the currently investigated models, justifying the choice of complex models like Random Forests.

* The motivation for considering multiple infusions is now clearly stated, but the reason for training separate models is not. The authors gave some arguments in their response to my comment 1, which I think could be added to the manuscript (e.g., to the Discussion).

* In Supplementary Figure S1, the total number of all patients is 949, although it should be 959. Maybe the right-most number should be 122 instead of 112?

* Despite the authors' reponse and the changes they have made to the manuscript, I still have some questions regarding the modelling process: (i) Supplementary Table S3 merely lists the final hyperparameter values, but not the set of possible values that were explored during Grid Search. Moreover, most of these values are obvious (e.g., num_class=3). (ii) Supplementary Table S3 does not list any hyperparameters of the Random Forest model. Does this mean the hyperparameters of the Random Forest were not tuned at all? (iii) The authors should clearly state in the manuscript that they tuned hyperparameters using Grid Search. Currently, the terms "grid search" and "tuning" do not appear in the revised manuscript.

* Thanks to the authors' response it is now clear that over- and undersampling were only applied in some kind of "secondary analysis", but not in the main analysis whose results are summarized in Table 2. This information should be added to Section "Splitting and generating datasets".

7. PLOS authors have the option to publish the peer review history of their article (what does this mean? ). If published, this will include your full peer review and any attached files.

**Do you want your identity to be public for this peer review?** For information about this choice, including consent withdrawal, please see our Privacy Policy .

Reviewer #1: **Yes: ** Amr Arafat

Reviewer #2: No

---

## [Author Response · Author response to Decision Letter 2]

16 May 2025

Reviewer #2:

1. The authors' response to the comment concerning the comparison to a simple baseline is not satisfactory. The authors claim that ML-models like Random Forests etc. better capture non-linear relationships (which is true, of course), but the main question is whether the data exhibit such non-linear relationships. I really want the authors to train simple baselines and include the results in the manuscript. Ideally, the baselines perform worse than the currently investigated models, justifying the choice of complex models like Random Forests.

Thank you for pointing this out. As requested, we trained a logistic regression model as a simple baseline using the sigmoid function. The methods and results are described in Table 2 and Supplementary Tables S3 and S9. The logistic regression model did not outperform the more complex models across any performance metrics under both first-dose and multiple-dose conditions. These findings suggest that the data likely contain non-linear relationships, supporting the use of more sophisticated models such as Random Forests. We have clarified this point in the Discussion section.

2. The motivation for considering multiple infusions is now clearly stated, but the reason for training separate models is not. The authors gave some arguments in their response to my comment 1, which I think could be added to the manuscript (e.g., to the Discussion).

Thank you for your important comments. We have compared several ML prediction models with different algorithmic features for optimal prediction with this data, which incorporates time series data including vital signs. We have described this in Discussion.

3. In Supplementary Figure S1, the total number of all patients is 949, although it should be 959. Maybe the right-most number should be 122 instead of 112?

We apologize for the incorrect description; the 112 in Supplementary FigureS1 is incorrect and the correct number is 122. The figures in Supplementary FigureS1 and the description have been corrected to the correct values.

4. Despite the authors' reponse and the changes they have made to the manuscript, I still have some questions regarding the modelling process: (i) Supplementary Table S3 merely lists the final hyperparameter values, but not the set of possible values that were explored during Grid Search. Moreover, most of these values are obvious (e.g., num_class=3). (ii) Supplementary Table S3 does not list any hyperparameters of the Random Forest model. Does this mean the hyperparameters of the Random Forest were not tuned at all? (iii) The authors should clearly state in the manuscript that they tuned hyperparameters using Grid Search. Currently, the terms "grid search" and "tuning" do not appear in the revised manuscript.

(i) Thank you for your suggestion. We searched hyperparameters by grid search only for multineural network. We did not tune hyperparameters for any other methods. We have listed the candidate sets searched and the selected hyperparameters in bold, and removed items that we thought were obvious from the text, such as num_class=3.

(ii) Thank you for your question. We do not adjust hyperparameters for random forests. Therefore, the hyperparameters are analyzed with default values. We have clearly stated this in the Supplementary Table S3.

(iii) Thank you. We have specified in the “Construction of prognostic model” of the method that the hyperparameters are tuned by grid search only for the multineural network.

5. Thanks to the authors' response it is now clear that over- and undersampling were only applied in some kind of "secondary analysis", but not in the main analysis whose results are summarized in Table 2. This information should be added to Section "Splitting and generating datasets".

Thank you for your suggestion. To avoid confusion, we have clarified in the “Splitting and generating datasets” section that over- and undersampling techniques were only applied in a secondary analysis using the best-performing model (based on ROC AUC), and not in the main analysis presented in Table 2.

---

## [Decision Letter · Decision Letter 2]

Analysis of aPTT predictors after unfractionated heparin administration in intensive care units using machine learning models

PONE-D-24-48061R2

Dear Dr. Kamio,

We’re pleased to inform you that your manuscript has been judged scientifically suitable for publication and will be formally accepted for publication once it meets all outstanding technical requirements.

Kind regards,

Elvan Wiyarta, M.D.

Academic Editor

PLOS ONE

Additional Editor Comments (optional):

Reviewers' comments:

Reviewer's Responses to Questions

**Comments to the Author**

1. If the authors have adequately addressed your comments raised in a previous round of review and you feel that this manuscript is now acceptable for publication, you may indicate that here to bypass the “Comments to the Author” section, enter your conflict of interest statement in the “Confidential to Editor” section, and submit your "Accept" recommendation.

Reviewer #1: All comments have been addressed

Reviewer #2: All comments have been addressed

2. Is the manuscript technically sound, and do the data support the conclusions?

Reviewer #1: Yes

Reviewer #2: Yes

3. Has the statistical analysis been performed appropriately and rigorously? 

Reviewer #1: Yes

Reviewer #2: Yes

4. Have the authors made all data underlying the findings in their manuscript fully available?

Reviewer #1: No

Reviewer #2: No

5. Is the manuscript presented in an intelligible fashion and written in standard English?

Reviewer #1: Yes

Reviewer #2: Yes

6. Review Comments to the Author

Reviewer #1: I would like to thank the atuhors for providing a revised version.

The author's response is satisfactory for me. I have no further comments.

Reviewer #2: (No Response)

7. PLOS authors have the option to publish the peer review history of their article (what does this mean? ). If published, this will include your full peer review and any attached files.

**Do you want your identity to be public for this peer review?** For information about this choice, including consent withdrawal, please see our Privacy Policy .

Reviewer #1: **Yes: ** Amr A. Arafat

Reviewer #2: No

---

## [Editor Report · Acceptance letter]

PONE-D-24-48061R2

PLOS ONE

Dear Dr. Kamio,

I'm pleased to inform you that your manuscript has been deemed suitable for publication in PLOS ONE. Congratulations! Your manuscript is now being handed over to our production team.

Kind regards,

on behalf of

Mr. Elvan Wiyarta

Academic Editor

PLOS ONE